# Unveiling the Prognostic Significance of BCL6+/CD10+ Mantle Cell Lymphoma: Meta-Analysis of Individual Patients and Systematic Review

**DOI:** 10.3390/ijms241210207

**Published:** 2023-06-16

**Authors:** Dani Ran Castillo, Daniel Park, Won Jin Jeon, Bowon Joung, Jae Lee, Chieh Yang, Bryan Pham, Christopher Hino, Esther Chong, Andrea Shields, Anthony Nguyen, Joel Brothers, Yan Liu, Ke K. Zhang, Huynh Cao

**Affiliations:** 1Department of Oncology/Hematology, School of Medicine, Loma Linda University, Loma Linda, CA 92354, USA; drcastillo@llu.edu (D.R.C.);; 2Department of Internal Medicine, School of Medicine, University of California San Francisco-Fresno, Fresno, CA 93701, USA; 3Department of Internal Medicine, School of Medicine, Loma Linda University, Loma Linda, CA 92354, USA; 4School of Medicine, Loma Linda University, Loma Linda, CA 92354, USA; 5Department of Internal Medicine, School of Medicine, University of California Riverside, Riverside, CA 92521, USA; 6Department of Pathology, Loma Linda University, Loma Linda, CA 92354, USA; 7Department of Nutrition, Texas A&M University, College Station, TX 77030, USA; 8Center for Epigenetics & Disease Prevention, Institute of Biosciences & Technology, College of Medicine, Texas A&M University, Houston, TX 77030, USA

**Keywords:** BCL6, CD10, aberrant MCL immunophenotype, overall survival

## Abstract

Mantle cell lymphoma (MCL) is a type of non-Hodgkin lymphoma (NHL) characterized by a hallmark translocation of t (11; 14). CD10 negativity has been used to differentiate MCL from other NHL types; however, recently, there has been an increase in the number of reported cases of CD10-positive MCL. This warrants further investigation into this rarer immunophenotype and its clinical significance. BCL6, which is a master transcription factor for the regulation of cell proliferation and key oncogene in B cell lymphomagenesis, has been reported to have co-expression with CD10 in MCL. The clinical significance of this aberrant antigen expression remains unknown. We conducted a systematic review by searching four databases and selected five retrospective analyses and five case series. Two survival analyses were conducted to determine if BCL6 positivity conferred a survival difference: 1. BCL6+ vs. BCL6− MCL. 2. BCL6+/CD10+ vs. BCL6−/CD10+ MCL. Correlation analysis was conducted to determine if BCL6 positivity correlated with the Ki67 proliferation index (PI). Overall survival (OS) rates were performed by the Kaplan–Meier method and log-rank test. Our analyses revealed that BCL6+ MCL had significantly shorter overall survival (median OS: 14 months vs. 43 months; *p* = 0.01), BCL6+/CD10+ MCL had an inferior outcome vs. BCL6+/CD10− MCL (median OS: 20 months vs. 55 months *p* = 0.1828), BCL6+ MCL had significantly higher percentages of Ki67% (Ki67% difference: 24.29; *p* = 0.0094), and BCL6 positivity had a positive correlation with CD10+ status with an odds ratio 5.11 (2.49, 10.46; *p* = 0.0000286). Our analysis showed that BCL6 expression is correlated with CD10 positivity in MCL, and BCL6 expression demonstrated an inferior overall survival. The higher Ki67 PI in BCL6+ MCL compared to BCL6− MCL further supports the idea that the BCL6+ immunophenotype may have prognostic value in MCL. MCL management should consider incorporating prognostic scoring systems adjusted for BCL6 expression. Targeted therapies against BCL6 may offer potential therapeutic options for managing MCL with aberrant immunophenotypes.

## 1. Introduction

Mantle cell lymphoma (MCL) is an NHL type that has a median overall survival (mOS) that can range from 1.8 to 9.4 years and comprises approximately 2.5–6% of B-cell NHL [1]. MCL is heterogeneous, with distinctive histological and molecular features, including a spectrum from indolent to aggressive types [2].

The classic genetic alteration in MCL is the chromosomal translocation of cyclin D1 and heavy-chain Ig at t (11; 14) (q13; q32). The constitutive dysregulation of cyclin D1 deregulates the cell cycle at the G1/S phase transition, facilitating neoplastic proliferation [3]. MCL is believed to arise from the pre-germinal center of naïve B cells; however, studies have shown that 15–30% of MCL have immunoglobulin-heavy variable somatic hypermutations, which suggests a possible origin in the germinal center [4]. CD10 protein expression is a hallmark of germinal center (GC) origin and is known for regulating the pro-neoplastic phosphatase and tensin homolog (PTEN) pathway [5]. MCL is typically thought to be CD10-negative; however, there has been an increase in the number of reported cases of CD10-positive MCL, suggesting a germinal center-derived pathology [6]. Figure 1 Illustrates the pathogenesis of MCL [7].

Studies have further reported a unique BCL6+/CD10+ MCL immunophenotype. BCL6 is a protein expressed in GC B cells, like CD10. This unique aberrant expression has primarily been described in a few case reports and small case series. These studies have characterized genetic alterations of the BCL6 gene and IGVH mutations, with a minority of reports detailing the presence of IGVH somatic hypermutations [8]. Thus, we propose that the expression of CD10 and/or BCL6 may provide prognostic value in newly diagnosed MCL, especially in patients with an atypical immunohistologic profile.

Patients with higher Ki-67% and blastoid morphology are often excluded in large population-scaled clinical trials in MCL due to lack of volume; therefore, an understanding of their association with aberrant immunophenotypes and clinical features will provide valuable insight for future study design, including novel agents and cell therapy [9].

### BCL6 and CD10 Relationship

The BCL6 gene is located on chromosome 3 (3q27) and encodes a POZ/Zinc finger transcriptional repressor protein that is largely restricted to GC B-cells [10]. It is required for GC formation and the T-cell-mediated immune response. In lymphoid malignancies, in addition to the constitutive BCL6 expression of GC B-cells, the gene can be deregulated by different mechanisms, including BCL6 rearrangement, somatic mutations in the 5′ noncoding region, and accumulated mutations in the regulatory region.

BCL6 is a transcriptional regulator that controls the expression of genes involved in various cellular processes. These include cell activation, differentiation, proliferation, apoptosis, and DNA damage response. In normal germinal centers (GCs), BCL6 is highly expressed in response to extracellular signals. As a result, the expression of IRF4 and PRDM1 is reduced, which leads to a highly proliferative and undifferentiated state in B-cells.

The specific genes regulated by BCL6 may vary depending on the cell type and context. Additionally, dysregulation of BCL6 has been implicated in various diseases, including lymphoma and autoimmune disorders [11].

CD10 expression is also closely associated with early lymphoid progenitors and GC cells [10]. However, its presence in a variety of other cell lines may imply a more diverse function than previously suggested. Various cases of CD10-positive and BCL6-positive MCL have been reported in the literature [7]. CD10 and BCL6 are typically labeled follicular center cell-associated antigens more commonly associated with other NHL types such as follicular lymphoma, diffuse large B-cell lymphoma (DLBCL), and Burkitt lymphomas [12]. Although these aberrant immunophenotypes of MCL are noted in the literature when totaled amongst other cases of MCL, BCL6 and CD10 positivity remains uncommonly co-expressed aberrant antigen expressions. Gualco et al., in an analysis of 127 MCL aberrant immunophenotypes, reported BCL6 positivity in 12% of cases without significant co-expression of CD10 [10]. Other analyses have suggested that the co-expression of CD10 and BCL6 markers in MCL is prone to a higher Ki-67 index. Although, it is notable that Pizzi et al. suggested that each marker was also independently associated with a higher Ki-67% [13].

Some studies have demonstrated the correlation of BCL6 and CD10 with MUM1 positivity in lymphoma [10,13]. Other studies have suggested a possible relationship between BCL6 and CD10 expression with apoptosis and cell cycle progression. The relationship is suspected to occur due to a potential correlation between BCL6, CD10, and cell proliferative associated proteins, such as cyclin A, high Ki-67%, and low BCL2 [14]. However, this study was conducted under the context of diffuse large B-cell lymphomas, highlighting the need for a greater understanding of the co-expression of BCL6 and CD10 in MCL. The mechanistic explanation of the interactions between BCL6 and CD10 positivity, along with these other biomarkers, remains unclear in current literature.

## 2. Methods

To design our search strategy, we used the PICOS (populations, interventions, comparators, outcomes, and study designs) model and followed PRISMA guidelines. We searched PubMed, EMBASE, and Cochrane Review to identify studies reporting CD10+ and BCL6+ mutations in MCL patients. We reviewed meta-analyses, retrospective studies, case series, and case reports.

The search terms included “mantle cell lymphoma”, “BCL6 mantle cell lymphoma”, “CD10 positive mantle cell lymphoma”, “Ki67% mantle cell lymphoma”, “mantle cell lymphoma mutation”, and “mantle cell lymphoma mortality”. Inclusion criteria included studies that reported BCL6 positive or negative MCL, CD10 positive or negative MCL, and Ki67% in BCL6 positive or negative MCL. BCL6 and CD10 negative patients served as a control group. Exclusion criteria included studies that focused on non-MCL types of non-Hodgkin lymphoma, lacked patient outcome data, did not specify individual survival months, or pertained only to animal model studies and cell line model studies.

Overall survival data from multiple studies used to collect the Ki67%data were pooled, and survival analysis was performed using the R programming package survival. The log-rank test was used to compare overall survival between the BCL6+ and BCL6− groups. Ki67% data from multiple studies were collected, and the mean differences between BCL6+ and BCL6− for each individual study were calculated using the inverse variance method and combined to minimize batch effects. For studies without individual Ki67 indexes, the standard deviation of the mean difference was calculated based on the Normal distribution and the sample size of the study. Summary statistics across all studies were calculated by averaging all mean differences and pooling individual standard deviations. Fisher’s exact test was used to test the correlation between CD10 and BCL6, and statistical significance was determined using a two-tailed alpha (α) level of 0.05.

## 3. Results and Discussion

### 3.1. Eligible Studies and Characteristics

Of the initial 125 articles identified through database screening, 93 were excluded after full-text screening for not meeting the predefined inclusion and exclusion criteria. The remaining 32 articles were chosen for further review. Ultimately, 10 articles were selected for data extraction and analysis. Screening of the Genomic Data Commons (GDC) and database of Genotypes and Phenotypes (dbGaP) yielded 0 patients meeting the inclusion criteria as there was no data found specific to mantle cell lymphoma. Figure 1 provides a flow diagram of the study screening and selection process. The aggregate patient characteristics of the 102 patients included in the analysis were as follows: CD10+: 70, CD10−: 467, and BCL6+/CD10+: 22. Data extracted from the articles included demographics, cytologic variants (classic, blastoid, pleomorphic, small), immunophenotype, overall survival months per individual, and Ki67%. Table 1 provides a cumulative summary of the extracted data points.

### 3.2. Clinicopathological Features of CD10 Positive and Negative MCL

The diagnosis of MCL has been confirmed using a combination of methods: IHC staining for cyclin D1 and/or SOX11, identification of the t (11; 14) (q13; q32) translocation by conventional cytogenetic analysis, or detection of CCND1 rearrangement by FISH studies.

In addition, the histology of involved tissue biopsies has been assessed and found to vary from classic (nodal, mantle zone, interstitial, and diffuse) to aggressive (blastoid/pleomorphic) forms.

### 3.3. Overall Survival BCL6+/BCL6−

Our analysis of the BCL6 expression effect (BCL6+ vs. BCL6−) on overall survival revealed that BCL6 positivity was associated with shorter median OS compared to patients with BCL6 negativity (14 months vs. 43 months; *p* = 0.01). The analysis demonstrated that aberrant expression of BCL6 was associated with worse outcomes compared to BCL6-negative patients. Figure 2 illustrates the Kaplan–Meier curve for BCL6+ vs. BCL6− MCL.

### 3.4. CD10+/BCL6+ vs. CD10+/BCL6−

Our correlation analysis showed that BCL6 positivity has an odds ratio of 5.11 (2.49, 10.46) for CD10 positivity compared to CD10 negative patients (*p* = 0.0000286). Furthermore, from analysis of 11 BCL6+/CD10+ patients and 6 BCL6−/CD10+ patients, patients with BCL6 and CD10 co-expression had shorter mOS compared to CD10+/BCL6− negative MCL (20 months vs. 55 months, *p* = 0.1828).

Thus far, our findings have characterized that: 1. BCL6 positivity is associated with poorer overall median survival vs. BCL6 negative MCL 2. BCL6+/CD10+ MCL has inferior mOS vs. BCL6−/CD10+ MCL 3. BCL6 has a positive correlation with CD10 positivity compared to CD10-negative patients. Overall, these findings contribute to a better understanding of the potential prognostic value of BCL6/CD10 expression in MCL.

### 3.5. Ki67% Proliferation Index

For the Ki67% analysis, 9 of the 10 studies selected from the literature review were utilized. The data for analysis and results are listed in Table 2. Overall, there was noted to be a 24.29 Ki67% difference between BCL6+ and BCL6− MCL (*p*-value: 0.0094). In the Chen et al. study, 63.13 +/− 19.45 was the average Ki67% of eight patients with CD10+/BCL6+ co-expression. Furthermore, 45+/− 21.69 was the average Ki67% of 87 patients without CD10+ and BCL6+ co-expression. In the Pizzi et al. study, 38 represented the mean Ki67% of 18 patients with BCL6 positive MCL, and 22 was the mean Ki67% of 148 patients with BCL6 negative MCL. For Fukushima et al., the Ki67% represents data from one patient with both BCL6+ and BCL6− MCL at different anatomic locations. The Forest plot analysis (Figure 3) of Ki67% in BCL6+ vs. BCL6− patients extracted from studies showed that BCL6+ expression correlated significantly with a higher mean proliferation index (Ki67%). Given our prior analyses resulted in worse median OS in BCL6 positive MCL, the 24.29 Ki67% difference may provide perspective as to the survival difference of BCL6+ vs. BCL6− MCL.

### 3.6. Discussion

#### 3.6.1. Immunohistologic and Morphologic Features of MCL

Conventionally, the majority of cases with classical MCL share a similar immunophenotypic profile, which is positive for CD5, cyclin D1 (less than 5% of cases are cyclin D1 negative), and SOX11 expression in the absence of follicular center cell-associated antigens [21]. Recent studies have characterized aberrant germinal-derived MCL immunophenotypes. These aberrant antigen expressions may be BCL6 and/or CD10 positive and lack CD5 expression. The expression of one of these immunophenotypes is not mutually exclusive, and unique co-expressions have been documented, such as the BCL6+/CD10+ immunophenotype. Given the heterogeneity of MCL immunophenotypes, a greater understanding of these unique antigen expressions is necessary to determine what prognostic value they hold and to determine what role they serve as diagnostic markers.

Morphology and immunophenotype are both significant factors in the heterogeneous response to MCL treatment. Typically, MCL exhibits a diffuse, non-nodular growth pattern of cells with irregular and cleaved nucleoli. This non-nodal MCL type (80–90%) usually has an unmutated immunoglobulin heavy chain region (IGHV) and lacks SOX 11 expression, indicating an indolent clinical course. Conversely, the blastoid or pleomorphic histological types of MCL (10–20%) have a more aggressive clinical course [22]. MCL cells are a homogenous population of medium-sized cells resembling lymphoblasts, while pleomorphic MCL cells are large anaplastic cells with irregular nuclei similar to those seen in DLBCL [9]. Immunophenotypically, blastoid/pleomorphic MCL is characterized by loss of CD5 expression, MYC aberrations, as well as somatic mutations in TP53, NOTCH1, and CCND1 [9,23]. In a study of CD10+ MCL with a large cohort (n = 30), Xu et al. reported a more diffuse growth pattern and higher rates of blastoid/pleomorphic morphology in CD10+ MCL patients (*p* < 0.0001). In the same study group, a higher rate of BCL6 expression was noted (36%, 6/19) in CD10+ MCL compared to CD10− MCL (7%, 5/72) (*p* = 0.009) [6]. In our study, 47.1% of CD10+ patients had classic morphology, while 48.6% had blastoid/pleomorphic/small morphology. For CD10− patients, 74.7% had classic morphology, and 16.8% had blastoid/pleomorphic/small morphology. Our study did not formally characterize morphology in relation to BCL6 positivity. Please refer to Table 1 for more details.

#### 3.6.2. The Immunophenotypic Significance of CD10 Positivity and BCL6 Positivity

Based on the current literature on CD10+ MCL, the significance of CD10 expression is controversial and mixed. Xu et al. and Akhter et al. were unable to demonstrate any difference in survival outcomes between CD10 positive and CD10 negative MCL [5,6]. Thus, CD10 expression alone does not appear to have a clear prognostic value in MCL. With this purpose, we conducted this analysis by combining multiple scientific studies in MCL aberrant antigen expressions to address the clinical importance of CD10 expression in MCL and further determine if BCL6 expression conferred a survival difference. Chen et al. found that patients with co-expression of CD10 and BCL6 had statistically significant lower one-year OS and event-free survival (EFS) (*p* = 0.011 and *p* = 0.015, respectively) [12]. Our analysis demonstrated that BCL6 expression may have superior prognostic value compared to CD10 expression, and BCL6 expression is positively correlated with CD10 expression. Furthermore, our analysis concluded that BCL6 expression was significantly associated with decreased survival in patients with MCL. As such, prognostic scoring systems adjusted for BCL6 expression could guide the management of MCL in specific populations.

CD10 expression may play a role in identifying the origin of various lymphomas from germinal center (GC) B cells. The co-expression of CD10 and BCL6 raised the possibility that these rare cases of MCL may arise from germinal center B cells. Care must be taken when approaching the diagnosis of patients with co-expression of CD10, MUM1, and BCL6, as this may lead to a misdiagnosis of follicular lymphoma (FL) with late GC phenotype in cases with a nodular growth pattern. Misdiagnosis of DLBCL and/or Burkitt’s lymphoma (BL) may also occur in specific cases with diffuse growth, given their pleomorphic/blastoid cytology. The positivity for MUM1 and BCL6 may point to the diagnoses of CD10-negative types of FL or B-cell chronic lymphocytic leukemia (CLL) [10,24]. Conversely, CD10 expression alone without BCL6 expression should prompt work to rule out phenotypically aberrant cases of other forms of B-cell malignancies, including hairy cell leukemia and nodal/extranodal marginal zone lymphoma [25].

The study conducted by Zanetto et al. aimed to investigate the relationship between CD10 expression in MCL and their derivation from GC B cells with somatic hypermutation of immunoglobulin genes. The study analyzed the sequences of the variable regions of the clonal rearranged IGH genes from five cases of MCLs and found that CD10+ MCLs are not different from other MCLs in terms of their IGVH mutation status. The results suggest that CD10 expression does not necessarily indicate a germinal center derivation for MCLs, contradicting previous studies that proposed CD10 expression as a potential marker for GC B cell derivation. However, the study also suggested that the gain of CD10 expression may occur during transformation to more aggressive types, indicating a potential role of CD10 in MCL progression [14].

The Chen et al. study identified 8 out of 104 MCL patients who exhibited co-expression of CD10 and BCL6. All eight patients were at stage 4, and six of them (75%) showed bone marrow involvement. Additionally, 75% (6/8) of the patients had elevated levels of LDH, and all eight had elevated levels of Beta-2 microglobulin [12]. LDH is one of the criteria used to calculate a MIPI score, and Beta-2 microglobulin has been reported by Yoo et al. as a possible independent and significant prognostic factor in patients with MCL [26]. The co-expression of the BCL6/CD10+ immunophenotype is unique, and only one other study by Camacho et al. was found in the literature discussing LDH and beta-2 microglobulin in BCL6+/CD10+ MCL. Among six patients with BCL6+ MCL, only one patient co-expressed BCL6+/CD10+ and had elevated Beta-2 microglobulin and LDH. Three out of the remaining five BCL6+ patients (60%) were noted to have elevated Beta-2 microglobulin and LDH, which is consistent with the study reported by Yoo et al. [8].

As research into this immunophenotype continues to increase, it may help to elucidate whether the BCL6+ status is an independent prognostic marker, irrespective of patient characteristics.

#### 3.6.3. Implications of Ki67% for MCL

Ki-67 is a nuclear protein associated with cellular proliferation as its expression is specifically correlated to cell cycle phases S, G1, G2, and M but is nonexistent in G0 [27]. The percentage of Ki-67-expressing cells in a tumor, known as the Ki-67 proliferation index (PI), defined by the percentage of Ki-67-positive lymphoma cells on histopathological slides, has since become an important predictive and prognostic marker in certain cancers [28,29]. Contrary to other B-cell NHLs, high Ki-67 PI has been demonstrated to be a poor prognostic factor in mantle cell lymphoma and has thus been standardized for routine evaluation on immunohistochemical analysis [30,31]. Cut-off values of Ki-67 PI > 40% have been correlated with a median survival of 15 months [30]. In a recent study by Jain et al., the Ki-67% was defined as low if < 30% and moderately high for 30–50% [32]. More recently, studies have shown that when Ki-67 PI is integrated with the Mantle Cell International Prognostic Index (termed MIPI-c), there are refined prognostic significance [33]. Certainly, the concept of combining MIPI with other biologic markers for risk stratification has been previously explored [34]. The survival difference in BCL6+ vs. BCL6− MCL is likely due to multiple factors, and the Ki67% difference may only provide some perspective. Further research is needed to fully understand the prognostic implications of BCL6 expression and Ki67% in MCL.

In a retrospective study from a single center experience by Chen et al., eight MCL patients with CD10 and BCL6 co-expression were primarily elderly men in stage IV, with a median age of 63. These patients had a higher Ki-67 index (63.13% ± 19.45%) and were more likely to present with aberrant immunophenotypes [12]. In contrast, another study conducted by Gao et al. found that only 8% of their MCL cases expressed CD10, and there was no significant difference in Ki-67 rate between MCL with an aberrant immunophenotype vs. MCL with a typical immunophenotype [35].

Our analysis indicates that BCL6+ expression was significantly correlated with higher Ki67% compared to BCL6− mantle cell lymphoma (*p* = 0.0094), which is consistent with the previous literature findings. This further demonstrates that the BCL6+ immunophenotype may provide valuable prognostic information.

The study highlights the importance of analyzing the expression of multiple markers, such as CD10 and BCL6, in MCL patients to gain a better understanding of their prognosis and determine the most effective treatment options. The results also suggest that the Ki-67 index can be used as a valuable prognostic indicator in MCL patients.

Small molecular inhibitors targeting BCL6 have the potential to be effective in treating certain types of cancers by disrupting the function of this protein, which plays a crucial role in the development and progression of these tumors. However, designing such inhibitors is a complex process that requires a thorough understanding of the structure and function of BCL6, as well as the specific sites involved in its tumorigenic activity. Overall, the development of BCL6 inhibitors represents a promising area of research in the field of cancer treatment, and combining these inhibitors with other targeted therapies may be an important direction for future research.

## 4. Conclusions

Our systematic analysis demonstrated that variations in immunophenotype are a rising phenomenon in mantle cell lymphoma. We have addressed heterogeneous findings about the prognostic value of CD10 expression in MCL. CD10 expression alone does not have clear prognostic value. However, there may be a significance in the expression of BCL6 in MCL. In this study, we found: 1. BCL6 positivity was associated with worse mOS vs. BCL6 negative MCL (median OS: 14 months vs. 43 months; *p* = 0.01) 2. BCL6+/CD10+ MCL has inferior mOS vs. BCL6−/CD10+ MCL (median OS: 20 months vs. 55 months; *p* = 0.1828) 3. A positive correlation between BCL6 and CD10 positivity with an odds ratio of 5.11 (2.49, 10.46; *p* = 0.0000286) 4. BCL6 positivity is associated with a higher Ki67% index vs. BCL6 negative MCL (Ki67% difference: 24.29; *p* = 0.0094). Awareness of the aberrant antigen expressions of MCL could afford clinicians foresight of the atypical pathophysiology of the disease, which may impact the management and associated clinical outcomes. Further research into BCL6+ immunophenotypes is necessary to more definitively characterize whether BCL6 positivity is prognostic independent of a patient’s underlying characteristics. This study concludes that greater investigation is required into the immunophenotypic and morphologic variants of MCL as a better understanding of this heterogenetic neoplastic process and the interplay between their cellular regulators will contribute to future precision medicine and targeted treatment in this heterogeneous malignancy.

### Limitations

Our analysis lacks a significant amount of data online available; is unable to conduct a complete molecular, histological, and pathological analysis; and is unable to collect the frequency of the presence of deletion 17p/TP53 mutation, which is well-known for a high-risk subset. Limited data from studies on BCL6 positivity and association with beta-2 microglobulin, LDH, and marrow involvement. This analysis was limited by the different staining protocols, varying specimen conditions, and different cut-off thresholds for immunohistochemistry interpretation. In addition, this study is subject to any biases or errors of the original investigators.

## Data Availability

This original research has not been submitted elsewhere, is not under review by another journal, and has not been published previously.

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
