# Peer review of "Unveiling the Prognostic Significance of BCL6+/CD10+ Mantle Cell Lymphoma: Meta-Analysis of Individual Patients and Systematic Review"

_ijms, 2023, doi:10.3390/ijms241210207_

Round 1

Reviewer 1 Report (Previous Reviewer 1)

fine with the changes. Thank you

Author Response

Response to reviewer

Thank you for your feedback.

The introduction has been edited and condensed to enhance conciseness.

We have made grammatical and syntactical adjustments throughout the paper, in addition to addressing the highlighted sections in the revisions. Certain paragraphs have been omitted to improve the overall organization and flow.

The term 'subtype' referring to MCL as a type of NHL has been corrected to 'type.'

To provide a more accurate classification, we have used the term 'aberrant antigen expression' when discussing CD10 and BCL6."

If you need any further assistance or have any other specific requests, please let me know!

Reviewer 2 Report (New Reviewer)

The authors did a nice investigation of the significance of mantle cell lymphoma (MCL) with aberrant expression of bcl-6 and/or CD10. However, the manuscript is too lengthy, particularly the Introduction section, and the information and data are not well organized. 

MCL is a specific type of non-Hodgkin lymphoma, not a subtype. CD10 positive MCL is not a subtype of MCL, but is considered an aberrant antigen expression. The same applies to bcl-6 expression.

 The Quality of English Language used in manuscript should be improved by professional Engligh editing.

In the 2nd line of the first paragraph of Discussion, there is a redundant CD5.

Author Response

Thank you for your feedback.

The introduction has been edited and condensed to enhance conciseness.

We have made grammatical and syntactical adjustments throughout the paper, in addition to addressing the highlighted sections in the revisions. Certain paragraphs have been omitted to improve the overall organization and flow.

The term 'subtype' referring to MCL as a type of NHL has been corrected to 'type.'

To provide a more accurate classification, we have used the term 'aberrant antigen expression' when discussing CD10 and BCL6."

If you need any further assistance or have any other specific requests, please let me know!

This manuscript is a resubmission of an earlier submission. The following is a list of the peer review reports and author responses from that submission.

Round 1

Reviewer 1 Report

Thank you for sharing this interessting manuscipt:

The systemic review and meta-analysis investigated the prognostic significance of BCL6+/CD10+ mantle cell lymphoma. The analysis revealed that BCL6 expression had a negative impact on overall survival, and BCL6 positivity was correlated with CD10 positivity, suggesting that incorporating BCL6 expression into prognostic scoring systems may aid in MCL management.

The inclusion criteria shall be clearer. Exclusion reasons to be clear in the PRISMA.

This meta-analysis is missing interesting confounding variables such as:

LDH, beat2-Microglobuin, lymph nodes, bone marrow involvement, MIBIscore…etc. It is difficult to collect all of these data from retrospective studies, but having some of these, well certainly bring more novel information. It is also will clarify if BCL6 status or rather the patients’ characteristics are determining the prognosis or may be confirm an association with BCL6 status. As it has been reported that BCL6 positive pts have higher LDH, advanced stages and high beta2-Microglubin, may looking into that in a larger cohort might help. The authors mentioned some patient’s characteristics which seems to be linked together as stage and age. However, having a forest plot summarizing these figures might be helpful.

Sample size: Depending on the research question and available data, the sample size may not be large enough to draw definitive conclusions as authors did in their conclusion which might be too extreme.

Heterogeneity: It's important to assess (or at least acknowledge it in limitations part) the degree of heterogeneity between studies, as this can impact the validity of the meta-analysis.

Please review the sentence: and 10 were selected for data extraction and analysis. Screening of the GDC and dbGaP databases yielded 0 patients meeting the inclusion criteria.

Please review:

-Table 1: BCL6+ and BCL6- numbers and if sum is not equal to 70/467, please highlight in table.

-Our analysis indicates that BCL6+ was significantly correlated with higher Ki67% compared to BCL6+ mantle cell lymphoma (p=0.0094),

Reviewer 2 Report

This paper was considerable clinical significance and an be acceptable.

Reviewer 3 Report

The authors report unveiling the prognostic significance of BCL6+/CD10+ mantle cell lymphoma. However, it is an analysis of a small number of cases from 10 articles and the results of the cases are also incomplete. Thus, there is no useful information in this study.